# Prophylactic Effect of Bovine Colostrum on Intestinal Microbiota and Behavior in Wild-Type and Zonulin Transgenic Mice

**DOI:** 10.3390/biomedicines11010091

**Published:** 2022-12-29

**Authors:** Birna Asbjornsdottir, Alba Miranda-Ribera, Maria Fiorentino, Takumi Konno, Murat Cetinbas, Jinggang Lan, Ruslan I. Sadreyev, Larus S. Gudmundsson, Magnus Gottfredsson, Bertrand Lauth, Bryndis Eva Birgisdottir, Alessio Fasano

**Affiliations:** 1Department of Pediatric Gastroenterology and Nutrition, Mucosal Immunology and Biology Research Center, Massachusetts General Hospital, Boston, MA 02152, USA; 2School of Health Sciences, Faculty of Medicine, University of Iceland, 101 Reykjavik, Iceland; 3Unit for Nutrition Research, Landspitali University Hospital and Faculty of Food Science and Nutrition, University of Iceland, 101 Reykjavik, Iceland; 4Department of Molecular Biology and Pathology, Massachusetts General Hospital, Boston, MA 02114, USA; 5School of Health Sciences, Faculty of Pharmaceutical Sciences, University of Iceland, 101 Reykjavik, Iceland; 6Department of Scientific Affairs, Landspitali University Hospital, 101 Reykjavik, Iceland; 7Department of Infectious Diseases, Landspitali University Hospital, 101 Reykjavik, Iceland; 8Department of Child and Adolescent Psychiatry, Landspitali University Hospital, 105 Reykjavik, Iceland; 9Department of Pediatrics, Harvard Medical School, Harvard University, Boston, MA 02114, USA

**Keywords:** bovine colostrum, microbiota–gut–brain axis, dysbiosis, zonulin, neuroinflammation, mental disorders, anxiolytic effect, eubiosis, oligosaccharide, short-chain fatty acids

## Abstract

The microbiota–gut–brain axis (MGBA) involves bidirectional communication between intestinal microbiota and the gastrointestinal (GI) tract, central nervous system (CNS), neuroendocrine/neuroimmune systems, hypothalamic–pituitary–adrenal (HPA) axis, and enteric nervous system (ENS). The intestinal microbiota can influence host physiology and pathology. Dysbiosis involves the loss of beneficial microbial input or signal, diversity, and expansion of pathobionts, which can lead to loss of barrier function and increased intestinal permeability (IP). Colostrum, the first milk from mammals after birth, is a natural source of nutrients and is rich in oligosaccharides, immunoglobulins, growth factors, and anti-microbial components. The aim of this study was to investigate if bovine colostrum (BC) administration might modulate intestinal microbiota and, in turn, behavior in two mouse models, wild-type (WT) and Zonulin transgenic (Ztm)—the latter of which is characterized by dysbiotic microbiota, increased intestinal permeability, and mild hyperactivity—and to compare with control mice. Bioinformatics analysis of the microbiome showed that consumption of BC was associated with increased taxonomy abundance (*p* = 0.001) and diversity (*p* = 0.004) of potentially beneficial species in WT mice and shifted dysbiotic microbial community towards eubiosis in Ztm mice (*p* = 0.001). BC induced an anxiolytic effect in WT female mice compared with WT female control mice (*p* = 0.0003), and it reduced anxiogenic behavior in Ztm female mice compared with WT female control mice (*p* = 0.001), as well as in Ztm male mice compared with WT BC male mice (*p* = 0.03). As evidenced in MGBA interactions, BC supplementation may well be applied for prophylactic approaches in the future. Further research is needed to explore human interdependencies between intestinal microbiota, including eubiosis and pathobionts, and neuroinflammation, and the potential value of BC for human use. The MGH Institutional Animal Care and Use Committee authorized the animal study (2013N000013).

## 1. Introduction

Over recent decades, the fields of neuroscience and microbiology have become more entwined [1,2,3,4]. The gut–brain axis (GBA) involves bidirectional communication between the gastrointestinal (GI) tract and the central nervous system (CNS), and its importance in maintaining host immune and metabolic homeostasis has long been appreciated [1,5,6,7,8]. Research has shown that the commensal microorganisms in the intestinal lumen can influence host physiology and pathology. Furthermore, a growing body of research has been exploring the intestinal microbiota’s possible role in neurobehavioral, neurodevelopmental, and mental and neurodegenerative disorders [9,10,11].

The more recent concept of the microbiota–gut–brain axis (MGBA), therefore, includes the role of the intestinal microbiota and the CNS, neuroendocrine system, neuroimmune system, the hypothalamic–pituitary–adrenal (HPA) axis, sympathetic and parasympathetic arms of the autonomic nervous system (ANS), and the enteric nervous system (ENS) [3,12,13]. Changes in the intestinal environment appear to impact brain function and behavior and play a role in the development of inflammatory diseases in the intestinal tract, as well as the brain [14].

An imbalance of the intestinal microbiota, i.e., intestinal dysbiosis [15], involves loss of beneficial microbial input or signal, decreased microbial diversity, and an expansion of pathogenic microbes (pathobionts) [16,17,18]. As a consequence, dysbiosis can lead to loss of barrier function and increased intestinal permeability (IP) [19,20]. Zonulin is a family of structurally and functionally related peptides whose archetype member is pre-haptoglobin-2 [21,22]. Zonulin has been found to reversibly regulate intestinal permeability by modulating intercellular tight junctions (TJs) between epithelial cells, such as in the small intestine. The release of zonulin from the intestinal mucosa is triggered by several environmental stimuli, including intestinal dysbiosis, which is thought to be one of the mechanisms by which the intestinal microbiota can trigger inflammation in the CNS [23,24,25].

The mucosal barrier is of physical, biochemical, and immune nature, and the microbiota can be considered as part of this system because of the mutual influence occurring between the host and the luminal microorganisms. Different species of the intestinal microbiota seem to be able to impair and/or promote, or even reconstruct, the intestinal barrier function [19,26,27,28,29]. Increased antigen trafficking through a dysfunctional intestinal barrier due to dysbiosis allows harmful substances from the intestinal lumen to enter the bloodstream [19,25], subsequently triggering immune cell activation that may lead to systemic inflammation [30,31,32,33,34]. Defects in the intestinal barrier function, including dysbiosis, have been found in several psychiatric disorders, such as autism spectrum disorder (ASD) [35,36,37], schizophrenia [35,38,39,40], and anxiety disorders [41,42,43], which have been associated with increased inflammation [40,44,45,46,47,48,49]. Furthermore, a dysfunctional MGBA associated with neuroinflammation has been reported in ASD and attention deficit hyperactivity disorder (ADHD) [12,50,51,52,53,54,55]. More importantly, neuroinflammatory responses do not solely occur in the presence of local insult but can develop when normal functioning of CNS is challenged by distally occurring pathological events [56].

Overall, research suggests that dysbiosis of the intestinal microbiota and increased intestinal permeability (commonly, but not exclusively, due to dysbiosis) can affect brain function, mental health, and behavior [3,16,31,35,57,58,59,60,61,62,63,64,65,66].

Research shows that diet can both modulate the intestinal microbiota for maintaining or boosting/building eubiosis, or in the direction of developing dysbiosis [67,68,69,70,71,72,73,74,75,76,77]. Currently, the best-studied environmental effectors in microbiome variation are diet and antibiotic treatment [67,73,78,79,80,81,82,83,84,85]. However, understanding intestinal microbiota active modulation through external exposome and its impact on behavior or disease propensity is still in its infancy [15].

Colostrum, the first milk that the mammal produces after birth, is a natural source of both macro and micro-nutrients [86,87]. It is rich in immunoglobulins, growth factors, and anti-microbial components [87,88,89,90,91]. Bovine colostrum (BC) is important for the nutritional and immunological support, growth, and development of the newborn calf [92]. There is also increasing evidence that BC may be of value for treating various medical conditions and ailments in children and adults [92,93,94]. Moreover, extensive data show colostrum may have value for preventing and treating microbial infections, e.g., working via the host’s immune function by inactivating the microbes [87,95,96,97,98,99,100,101,102,103,104,105,106,107,108,109,110,111].

Identifying and researching the MGBA can be done through interventions investigating effects on microbiota and behavior [1,65]. Although we do not yet fully understand the functional significance of the symbiotic relationship between the host and the microbiome, especially in the context of brain health, several tools and animal models have been invaluable in allowing researchers to constantly narrow the gaps in understanding of the MGBA. One of these is the zonulin-transgenic mouse (Ztm) model, which is characterized by increased intestinal permeability, dysbiotic intestinal microbiota, activation of the innate immune system, and mild hyperactivity [27,39]. A better understanding of the mechanism of the MGBA involving the neuro-immune system might offer a new approach to the treatment of mental disorders. Therefore, targeting the microbiota with nutritional and therapeutic strategies could be a novel approach for improved brain health and well-being.

Due to dysbiosis and increased trafficking of pro-inflammatory bacterial species and their products into the systemic circulation, the Ztm are predisposed to inflammation, including neuroinflammation. The aim of this study is to investigate if BC application modulates the intestinal microbiota and behavior in two mouse models, i.e., wild-type (WT) and Ztm, compared with control mice (WT Ctr and Ztm Ctr). This is the first study investigating the potential effects of BC using the unique zonulin-expressing mouse.

## 2. Materials and Methods

### 2.1. Animals

A colony of C57Bl/6 WT mice was maintained at the Massachusetts General Hospital animal facility, and the zonulin transgenic mouse (Ztm) model was generated as reported previously [112]. Both colonies were housed in separate cages within the same facility for the duration of the study. After weaning at 4 weeks, both colonies were housed, raised, and maintained under standard settings, i.e., in a room with a 12 h light/dark cycle, standard temperature and humidity, and with ad libitum access to standard mice chow and water. Every effort was made to limit animal suffering and to utilize only the minimum number of animals required to obtain accurate results. The MGH Institutional Animal Care and Use Committee authorized the animal study (protocol code 2013N000013; 3 March 2021).

### 2.2. Experiment

We investigated if BC administration in drinking water ad libitum would modulate/affect: (1) microbiota in WT and Ztm and/or increase intestinal microbial eubiosis in Ztm mice and (2) behavior in WT and Ztm and/or ameliorate behavioral changes previously reported in Ztm mice [39].

For these experiments, 27 (11 females and 16 males) WT and 26 (11 females and 15 males) Ztm, were used (*n* = 53). Before the experiment, the animals were divided into four groups. Two groups of each genotype (WT and Ztm) were fed with fresh, unprocessed BC in their drinking water at a ratio of 1:1 *v*/*v* ad libitum. The remaining groups (control mice) had no BC added to their drinking water. All groups received regular chow and drinking fluid ad libitum throughout the duration of the experiment. The mice were weighed thrice weekly, and their fluid intake was thoroughly recorded daily. After four weeks, fresh stool samples from all mice were obtained for microbiota investigation to compare between all groups and to baseline from our previous experiments [27,39]. To assess anxiety-like, obsessive, compulsive, and repetitive-like behaviors, we performed behavioral assays, i.e., marble burying (MB) and the light-dark box (LDB) test, as previously reported [39,113,114,115,116,117,118]. The experimental timeline and flowchart are shown in Figure 1.

### 2.3. Bovine Colostrum (BC) Administration

Samples of BC were collected from authorized dairy farms in Iceland using robots, i.e., milking servers, and standard procedures were followed for the collection. BC was only collected from healthy dairy cows, and standard tests were performed to rule out infection or mastitis. Each farmer was required to register readings from the milking system and record information about the milking. A workbook was kept for methodology and frequency of cleaning and disinfecting the teats.

The first milking BC was collected in a sterile container, filtered, and refrigerated immediately at 4 °C. Then, the BC was transferred to one-liter sterile plastic bottles, selected in line with regulations regarding materials and objects intended to come in contact with food, within 12 h and placed immediately in a freezer at −20 °C until exported on dry ice (−80 °C) to the Mucosal Immunology and Biology Research Center at Massachusetts General Hospital.

During the experiment, an a priori calculated amount of BC was thawed daily and mixed with drinking water provided by the animal facility in a 1:1 *v*/*v*. The feeding took place at the same time every day, allowing for 2–3 h max flexibility. Water and BC were mixed thoroughly. To secure freshness, the BC was replaced daily, and all feeding bottles were cleaned thoroughly each time. The BC was well tolerated and did not induce any clinical symptoms in the animals.

Analyzing the raw material revealed the following per kg: water 80.7%, protein 9.8%, ash 1.1%, total fat 5.1%, and carbohydrates 4.1%, and the following minerals (in mg/kg ±20%): selenium 0.102, iron 1.33, copper 0.11, zinc 13.18, natrium 580, potassium 1.450, phosphorus 1.590, magnesium 290, chromium <100, and manganese 30.

### 2.4. Behavioral Assays

Two standard behavioral assays, MB [117,118] and the LDB [113,114,115,116] test (clarification in Section 2.4.1 and Section 2.4.2), were applied to investigate behavioral differences among the experimental groups. Behavioral evaluations were conducted when the mice were 8 weeks old (on day 28 of the experiment), over 4 days between 9:00 AM and 6:00 PM. To allow for behavioral habituation, the mice were brought into the experimental room at least thirty minutes prior to the start of the experiment. Males and females were never present in the testing room simultaneously and were always tested and acclimatized separately.

#### 2.4.1. Marble Burying (MB)

When mice are put in a cage with marbles, they will bury the marbles. As a paradigm of the ethologically natural behavior of defensive burying, the MB test examines obsessive/compulsive/repetitive behavior [117,118]. The experimental mice had not been previously exposed to glass marbles, leading to the reasonable conclusion that the novelty of the marble triggers the response of marble burying. Standard housing cages (28 × 20 × 12 cm) were prepared in the standard way, filled with 7 cm of autoclaved and evenly distributed bedding material (SANI-CHIP^®^) without nesting material. Twenty marbles (Fisher Science Education^TM^ glass marbles #S04581, 1.42 cm) were equally spaced in four parallel lines with alternating black and blue colors. Cages were transferred to the testing faculty area, females and males separately, and one mouse was placed in the center of each cage, where it could freely interact with the marbles. After 20 min, the mice were carefully removed to avoid disturbing the bedding and/or marbles, and the number of buried marbles was counted. Marbles were considered buried when at least 2/3 of their volume was covered by bedding material.

#### 2.4.2. Light Dark Box (LDB) Test

The LDB test may be useful to predict anxiolytic-like or anxiogenic-like activity in mice [116,119]. The LDB apparatus consists of two compartments, i.e., the light compartment makes up 2/3 of the box and is brightly lit and open, and the dark compartment comprises 1/3 of the entire box and is covered and dark. A 7 cm-wide door connects the 2 compartments of 35 cm-tall walls and is split into two 20 cm × 40 cm chambers. Rodents, including mice, prefer darker areas to lighter areas [113,114,115,116]. However, when in a new environment, the mice tend to explore. These two conflicting feelings lead to observable anxiety-like symptoms. No prior training is required for the LDB. There is no food or water deprivation, and only natural stressors, such as light, are used. Furthermore, an infrared beam sensor is attached to the ceiling allowing movement detection.

The mice were placed in the dark section of the unit and allowed to move around. Generally, the mice move around the perimeter of the compartment until they find the door. This process may take between 7 and 12 s. All four paws must be placed in the opposite compartment to be considered an entry. The distance each animal traveled in the light section, the total number of transitions, the time spent in each section, and the latency to enter the light section were recorded with Ethovision XT, an automated detection and quantification software. After each trial, all chambers were cleaned with super hypochlorous water to prevent a bias based on olfactory cues.

### 2.5. Microbiome Analysis; Sample Collection and DNA Extraction

Fresh fecal samples were collected from each individual mouse (*n* = 53) at eight weeks, after the behavioral tests were performed. The samples were flash-frozen and kept at −80 °C. Genomic DNA was extracted from each sample by applying the DNeasy powersoil extraction kit (Qiagen, Beverly, MA 01915, USA), following Qiagen’s instructions. To quantify the amount of DNA, Nanodrop was applied. Phylogenetic profiling was conducted by amplifying the hypervariable V4 region of the 16S rRNA gene by PCR, applying 5× prime master mix (MGH primer bank, Boston, MA 02114, USA). Reverse 806 primers were barcoded, and a unique forward 515 primer (Integrated DNA Technologies) was applied. Regular gel electrophoresis was run to confirm the correct amplification of the V4 regions. A QIAquick PCR purification kit (Qiagen, USA) was used for purifying PCR products. Concentration was measured by a Quant-iT Picogreen dsDNA kit, following the manufacturer’s instructions. The MGH NextGen Sequencing Core facility (Boston, USA) performed the sequencing of all samples, applying the Illumina system using the MiSeq v2 500 cycles reagent kit, following the manufacturer’s instructions. The system sequenced a total of 250 paired-end cycles for maximum coverage of the amplicon. The following primers were applied for the sequencing [120]:

read 1 (TATGGTAATT GT GTGYCAGCMGCCGCGGTAA)

read 2 (AGTCAGCCAGCCGGACTACNVGGGTWTCTAAT)

index (AATGATACGGCGACCACCGAGATCTACACGCT)

### 2.6. Bioinformatic Analysis of Microbiome Data

QIIME2 software package version 2018.2.0 was applied to process and analyze sequencing data [121]. Sequencing reads with low-quality scores (average Q < 25) were truncated to 240 bp and then filtered, applying the deblur algorithm (default settings) [122], and remaining high-quality reads were aligned to the reference library, applying *mafft* [123]. Next, the aligned sequences were masked to remove highly variable positions, and a phylogenetic tree was generated from the masked alignment by FastTree plugin [124]. Using default QIIME2 plugins [121], alpha and beta diversity metrics, and Principal Component Analysis, plots were generated [125,126,127,128,129,130]. To assign taxonomies to our sequences, we have used QIIME2’s feature-classifier plugin and pre-trained the Naïve Bayes classifier, which has been trained on the Silva 138 99% operational taxonomic units (OTUs) [131,132,133]. Differential abundance analysis of OTUs was performed using ANCOM [134].

### 2.7. Statistical Processing

Statistical analysis of the microbiome was performed, applying the Kruskal–Wallis test to assess statistical significance of abundance differences between groups. Alpha diversity analysis was performed with Kruskal–Wallis Rank-Sum and pairwise tests. For beta diversity analysis, permutation multivariate analysis of variance (PERMANOVA) with 1000 permutations was applied to test the significance of the community composition and structural differences among the groups. The Benjamini–Hochberg false discovery rate (FDR) was employed for multiple testing corrections, with a cutoff set at 0.05 (reported as q values) [135]. Statistical analyses for the behavioral experiments were performed in Graph Pad Prism 9, applying two-way ANOVA, and post-hoc analyses were carried out using Tukey’s HSD. Differences were considered statistically significant with *p* < 0.05.

## 3. Results

To investigate the modulating effect of colostrum on the intestinal microbial community, we analyzed the microbiome of eight-week-old WT and Ztm mice (*n* = 53). The composition of the intestinal microbiota of four-week-old mice is established, but still susceptible to fluctuations due to external factors, such as diet [136]. Therefore, this period (4–8 weeks) represents an excellent opportunity to influence the composition of the microbiota.

### 3.1. Alpha Diversity Analysis in Wild-Type (WT) and Zonulin Transgenic Mice (Ztm)

Microbial community structure can be assessed by alpha diversity, which includes measures of evenness (whether all species have a similar abundance within the community) and richness (the number of different bacterial species present). Alpha diversity was quantified using Pielou’s evenness, species richness, and Shannon diversity indexes, which relate OTU evenness and richness, the total number of observed species, and variation and complexity within the group. Alpha diversity analyses were performed with Kruskal–Wallis Rank-Sum and pairwise tests.

#### 3.1.1. Wild-Type (WT) Groups

Evaluation of the relative abundance of different species of intestinal microbiota in WT female and male mice showed that the alpha diversity calculated using Pielou´s evenness index was significantly different between the groups (*p* = 0.001), applying the Kruskal–Wallis Rank-Sum test (Figure 2a). There was a significantly increased relative abundance in one of the subgroups, i.e., in WT *f* BC vs. WT *f* Ctr (Kruskal–Wallis pairwise (*p* = 0.04; Table 1). The significant difference between the rest of the subgroups is due to different baselines between WT female and male mice [27,39]. Evaluation of diversity in each WT sample, including the number of different species, using the species richness index, showed no significant difference between the groups (Kruskal–Wallis Rank-Sum *p* = 0.2; Figure 2b), nor any of the subgroups (Table 1).

As indicated by the Shannon index, there was a significant difference in the diversity of microbial community abundances between the WT female and male groups (Kruskal–Wallis Rank-Sum *p* = 0.004; Figure 2c). When divided by sex, there was a significant difference between the subgroups, i.e., WT *f* BC vs. WT *f* Ctr (Kruskal–Wallis pairwise *p* = 0.04). Per contra, there was no difference between the WT male BC group when compared with the WT male control group (Kruskal–Wallis pairwise *p* = 0.1; Table 1).

Overall, a significant difference in the intestinal microbial community was observed in the WT female BC group when compared with the WT female control group. These findings suggest an increase in taxonomic abundance and diversity in the intestinal microbiota in WT female mice that underwent BC treatment, when compared with control mice. 

#### 3.1.2. Zonulin Transgenic Mice (Ztm) Groups

Evaluation of the relative abundance of different species’ intestinal microbiota in the Ztm female and male groups showed that, overall, the alpha diversity calculated using the evenness index was not significantly different in the BC groups when compared with the control groups (Kruskal–Wallis Rank-Sum *p* = 0.07; Figure 3a). No significant difference was found between the subgroups, i.e., Ztm *f* BC vs. Ztm *f* Ctr, nor Ztm *m* BC vs. Ztm *m* Ctr (Table 2).

Evaluation of the diversity in each sample, including the number of different species, using the species richness index showed significant difference between the Ztm treatment and control groups (Kruskal–Wallis Rank-Sum *p* = 0.001; Figure 3b), and in the subgroups, significant difference between Ztm *f* BC vs. Ztm *f* Ctr (Kruskal–Wallis pairwise *p* = 0.01) and between Ztm *m* BC vs. Ztm *m* Ctr (Kruskal–Wallis pairwise *p* = 0.002; Table 2). 

As indicated by the Shannon index, there was a significant difference in microbial community diversity between the Ztm groups (Kruskal–Wallis Rank-Sum *p* = 0.05; Figure 3c). However, there was no significant difference between the subgroups, i.e., Ztm *f* BC vs. Ztm *f* Ctr (Kruskal–Wallis pairwise *p* = 0.5), nor Ztm *m* BC vs. Ztm *m* Ctr (Kruskal–Wallis pairwise *p* = 0.08; Table 2).

Overall, a significant difference in the intestinal microbial community was observed between the Ztm treatment and control groups, as well as between the female and male subgroups (Ztm *f* BC vs. Ztm *f* Ctr), suggesting an increase in taxonomic richness and diversity characterizes the intestinal microbiota in Ztm female and male mice that received BC.

### 3.2. Beta Diversity Analysis

Beta diversity (the degree of pair-wise similarity in species composition among the groups) was assessed by applying Bray–Curtis and Jaccard indices. Similarity measures of abundance, presence, and absence data at the level of species, genera, families, orders, classes, and phyla were assessed in the four groups of mice (*f* & *m*), i.e., WT BC, WT Ctr, Ztm BC, and Ztm Ctr (Table 3; Figure 4).

PERMANOVA tests demonstrated significant differences in intestinal microbiota between the groups and within the groups. The Bray–Curtis beta diversity index showed the overall dissimilarity of bacterial communities was significantly different between the WT BC groups compared with the WT control groups (*p* = 0.03; Table 3) and between Ztm BC groups compared with Ztm control groups (*p* = 0.001; Table 3). Within the groups, there was a significant difference within the WT male group, i.e., WT *m* BC vs. WT *m* Ctr (*p* = 0.03; Table 3) and within the Ztm female and male treatment groups, i.e., Ztm *f* BC vs. Ztm *f* Ctr (*p* = 0.01; Table 3) and Ztm *m* BC vs. Ztm *m* Ctr (*p* = 0.001; Table 3).

Similarity in species composition between the groups assessed by Jaccard indices demonstrates significant differences between all WT groups (*p* = 0.04; Table 3) and within all groups. In the subgroups, i.e., WT female and male BC groups compared with control groups, differences in the abundance and presence/absence of taxa were observed, indicating that the beta diversity is significant (WT *f*, *p* = 0.003; WT *m, p* = 0.02; Table 3; Figure 4a,b). Within the Ztm groups, similar differences were found in the abundance and presence/absence of taxa, indicating beta diversity significance between the groups (*p* = 0.001; Table 3). In the subgroups, i.e., the Ztm female and male BC groups, the abundance and presence/absence were significantly higher than in the control groups, i.e., Ztm *f*, *p* = 0.004; Ztm *m*, *p* = 0.001 (Table 3; Figure 4c,d).

Furthermore, the relationships between groups of microbiota samples were assessed using principal component analysis (PCA), based on the Jaccard beta diversity index, as a measure of the overall dissimilarity of bacterial communities within and between the groups (Figure 5a–d). PCA of the intestinal microbial communities revealed a trend for partially overlapping sample groups in the WT (BC vs. Ctr) microbiome and distinct clustering of Ztm (BC vs. Ctr) microbiome profiles.

### 3.3. Species Composition

#### 3.3.1. Phylum Level

At the phylum level, Firmicutes and Bacteroidetes prevailed in all groups, i.e., Firmicutes 51.5% (WT *f* Ctr) and 49.4% (WT *m* Ctr), and 50.3% (Ztm *f* Ctr) and 42.9% (Ztm *m* Ctr), and Bacteroidota were 47.7% (WT *f* Ctr) and 49.2% (WT *m* Ctr), and 42.6% (Ztm *f* Ctr) and 47.9% (Ztm *m* Ctr) (Figure 6a,b).

However, there was a shift in Firmicutes abundance in all the BC groups when compared with the control groups. The WT female BC group harbored an increased abundance (56.8%), the WT male BC group harbored a decreased abundance (43.8%), and the Ztm female BC group harbored a decreased abundance (43.5%) when compared with the control groups (51.5%; 49.4%; 50.3%). The Ztm male BC group harbored an increased abundance (43.4%) compared with the control group (42.9%). Furthermore, there was a shift in Bacteroidota within the BC groups when compared with the control groups. The WT female BC group harbored a decreased abundance (40.1%) and the WT male BC group harbored an increased abundance (54.4%) compared with the control groups (47.7%; 49.2%). However, the Ztm female and the Ztm male BC groups harbored an increased abundance of (51.5%) and (49.6%) compared with the control groups (42.6% and 47.9%) (Appendix A).

Moreover, the WT BC groups harbored a decreased abundances of Proteobacteria compared with the control groups, but the Ztm BC groups harbored an increased abundance compared with the Ztm control groups. All BC groups harbored a decreased abundance of Actionbacteriota compared with the control groups, and all the BC groups, except one (Ztm *f* BC), harbored an increased abundance of Verrucomicrobiota. Campylobacterota was only detected in the Ztm groups, and the relative abundance was increased in the Ztm female BC group, but decreased in the Ztm male BC group compared with the control groups (Figure 6a,b; Appendix A). Deferribacterota and Desulfobacterota were only found in the Ztm groups, and both BC groups harbored a reduced relative abundance, i.e., females and males. Cyanobacteriota was only detected in the Ztm groups and was increased in the Ztm female BC group, but reduced in the Ztm male BC group (Appendix A).

Overall, there was a shift in all phyla in all treatment groups.

#### 3.3.2. Family Level

At the family level, the intestinal microbiota of all groups contained high levels of Muribaculaceae, Lachnospiraceae, and Erysipelotrichaceae (Figure 6c,d; Appendix A). However, there was a shift within the BC groups, and each comparison exhibited a different signature. The WT female BC group harbored a reduced abundance of Muribaculaceae (37.4%) compared with the WT female control group (44.7%), and the WT male BC group harbored an increased abundance (54.3%) compared with the WT male control group (49.2%). In the Ztm BC groups, the Ztm females harbored an increased abundance (38.6%) and the Ztm male BC group harbored an increased abundance (42.2%) compared with the Ztm control groups (34 % and 41.5 %), respectively (Appendix A).

Within the treatment groups, the WT groups harbored an increased abundance of Lachnospiraceae (WT *f* BC = 29.7% and WT *m* BC = 19.5%) compared with the control groups (22.4% and 13.6%), respectively. The Ztm treatment groups harbored a reduced abundance (Ztm *f* BC = 14.8% and Ztm *m* BC = 10.9%) compared with the control groups (19% and 15.3%), respectively. Decreased abundance of Erysipelotrichaceae was found in all the treatment groups (WT *f* BC = 8.0% and WT *m* BC = 7.9%, and Ztm *f* BC = 13.9%), except for the Ztm male BC group, as they harbored an increased abundance (21.6%) compared with the control groups (WT *f* Ctr = 14.8%; WT *m* Ctr =17.6%; Ztm *f* Ctr = 14.7%; Ztm *m* Ctr 15.6%), respectively (Appendix A).

The WT and Ztm female BC groups harbored a reduced abundance of Lactobacillaceae (0.6% and 8.3%) compared with the WT and Ztm female control groups (3.2% and 9.5%), respectively. However, the WT and Ztm male BC groups harbored an increased abundance (2% and 4.8%) when compared with the WT and Ztm male control groups (1.6% and 4%), respectively (Appendix A). Clostridiaceae abundance was reduced in both WT BC groups (0.5% and 0.4%) compared with the WT control groups (1% and 4.8%), and in the Ztm BC groups (0.04% and 0.05%) when compared with the Ztm control groups (0.1% and 0.1%). All BC groups harbored an increased abundance of Oscillospiraceae when compared with the control groups.

Akkermansiaceae was only found in the WT groups, and both WT BC groups harbored an increased abundance (2.9% and 0.8%) compared with the control groups (0.4% and 0.2%). There was a slight expansion in the Ztm male BC group (0.07%) compared with the Ztm male control group (0%). (Appendix A).

Clostridia UCG-014 relative abundance was increased in the WT female BC group (2.4%), in the Ztm female BC group (0.06%), and in the Ztm male BC group (0.2%), when compared with the control groups (1.1%; 0.05%; 0.1%). However, the relative abundance was reduced in the WT male BC group (2.7%) when compared with the control group (3.3%) (Appendix A).

Ruminococcaceae abundance was increased in the WT female BC group (3.3%), in the WT male BC group (3.2%), and in the Ztm male BC group (0.9%), when compared with the control groups (1%; 2.1%; 0.8%). However, the Ztm female BC group harbored a reduced abundance (0.4%) compared with the control group (1.6%) (Appendix A).

Overall, there was a shift in the microbial abundance in all bacterial families within all treatment groups, except for Prevotellaceae in the WT groups, Bacteriodaceae in the WT male group, and Akkermansiaceae in the Ztm female group.

#### 3.3.3. Genus Level

Volcano plot analysis showed significant differences between all the BC groups compared with the control groups (Figure 7a–d). The WT female BC group harbored a significantly increased abundance of *Eubacterium* (Ruminococcaceae family) and a significantly reduced abundance of *Tyzzerella* (Lachnospiraceae family) and *Dubosiella* (Erysipelotrichaceae family) *(*Figure 7a).

The WT male BC group harbored a significantly increased abundance of *Lactococcus* (Streptococcaceae family) and a significantly reduced abundance of *Clostridium sensu stricto* 1 (Clostridiaceae family) (Figure 7b). The Ztm female BC group harbored a significantly increased abundance of *Coriobacteriaceae* UCG-002 (Atopobiaceae family) and a reduced abundance *of Peptococcus* (Peptococcaceae family) (Figure 7c). The Ztm male BC group harbored a significantly reduced abundance of *Lachnospiraceae* UCG-001 (order Clostridia), *Desulfovibrio* (Desulfovibrionacea family), and *Mucispirillum* (Deferribacteraceae family), but an increased abundance of UCG-010 (order Oscillospirales) (Figure 7d).

At the genus level, the intestinal microbial communities of all groups harbored a high abundance of *Lachnospiraceae* NK4A136_group (Appendix A). By contrast, the WT groups (BC and Ctr) were characterized by a higher abundance of *Turibacter* compared with the Ztm groups (BC and Ctr), and the Ztm groups (BC and Ctr) were characterized by a higher abundance of *Faecalbacaculum*, *Dubosiella*, and *Parasuttella* compared with the WT groups (BC and Ctr). Both female groups, i.e., WT *f* (BC and Ctr) and Ztm *f* (BC and Ctr), were characterized by a higher abundance of *Roseburia* compared with the male groups (BC and Ctr; Appendix A).

The WT female BC group harbored an increased abundance of the *Ruminococcus* and *Eubacterium Xylanophilum* groups when compared with the WT control group. The WT male BC group harbored an increased abundance of *Oscillibacter, Eubacterium Xylanophilum* group A2 when compared with the WT male control group (Appendix A).

The Ztm female BC group harbored a reduced abundance of *Turicibacter, Bifidobacterium, Desulfovibrio*, *Incertae Sedis*, *Rombutsia,* and *Blautia* when compared with the Ztm female control group. In the Ztm male BC group, *Dubosiella*, *Clostridia* UCG-014, and *Ruminococcus* abundance were increased, but *Roseburia, Helicobacter, Bifidobacterium,* and *Desulfovibrio* abundance were reduced compared with the Ztm male control group (Appendix A).

In the total of 69 genera, there was a shift in the abundance of 44 genera within the WT treatment groups and of 61 genera withing the Ztm treatment groups.

#### 3.3.4. Analysis of Differential Species Abundances

A deeper comparison of the microbiota using the ANCOM method [134] confirmed a significantly increased abundance of the *Eubacterium siraeum* group; uncultured bacterium, *Acutalibacter muris* sp. (Rumnococcaceae family) and a significant reduction of *Tyzzerella*; uncultured bacterium (Lachnospiraceae family) and *Dubosiella*; uncultured bacterium (Erysipelotrichaceae family) in the WT female BC group compared with the WT female control group (Figure 6c and Figure 7a). There was a significantly increased abundance of uncultured species of the *Lactococcus* genus (Streptococcaceae family) and a significantly reduced abundance of uncultured bacterium (Peptococcaceae family) and *Clostridium sensu stricto* 1 sp. (Clostridiaceae family) in the WT male BC group when compared with the WT male control group (Figure 7b). In the Ztm female BC group, a significantly reduced abundance of Coriobacteriaceae_UCG-002 and *Peptococcus* genera (Peptococcaceae family) were found compared with the Ztm female control group (Figure 7c). In the Ztm male BC group, a significant reduction of the genera Lachnospiraceae_UCG-001, *Desulfovibrio* (Desulfovibrionaceae family), and *Mucispirillum schaedleri* sp. (Deferribacteraceae family), and a significantly increased abundance of UCG-010 genus (Ruminococcaceae family) was found compared to the Ztm male control group (Figure 7d).

Overall, there was an increase of species abundance in all treatment groups, except in the Ztm female group, where there was only a reduction.

### 3.4. Behavioral Assays

To assess behavioral patterns in the studied animals, we tested for repetitive and anxiogenic behavior in all groups by applying both MB and LDB tests (Figure 8). No significant difference was observed between any of the groups for the number of marbles buried (Figure 8a,b).

In the LDB test, there was no significant difference between any of the groups for time spent in open area (Figure 8c,d) or the total distance each animal traveled in the light section (Figure 8e,f).

Significant differences were found in the mean value of latency to first enter the light section (*Interaction effect* (F(1,15) = 8.8, *p* = 0.009)) and (*Treatment effect* (F(1,15) = 26.8, *p* = 0.0001))) in both WT and Ztm female BC groups when compared with the WT female control group (Figure 8g). Post hoc testing revealed significantly reduced latency to enter the light section, i.e., WT *f* BC vs. WT *f* Ctr (*p* = 0.0003, 95% C.I. = 0.57, 25.32), and Ztm *f* BC vs. WT *f* Ctr (*p* = 0.001, 95% C.I. = 7.56, 32.27).

The mean value of latency to first enter the light section was significantly different between the WT *m* BC group and the Ztm *m* BC group (*Genetic background effect* (F(1,25)) = 5.2, *p* = 0.03))), and post hoc testing revealed a reduced latency to enter the light section (*p* = 0.03, 95% C.I. = 11.31, 36.02; Figure 8h).

A significant difference was observed in the frequency of transitions between the WT female BC group and the Ztm female BC group (*Interaction effect* (F(1,14) = 5.4, *p* = 0.03; and *Genetic Background effect* (F(1,14) = 4.9, *p* = 0.04)). Post-hock analysis revealed increased transitions, i.e., WT *f* BC compared with Ztm *f* BC (*p* = 0.01, 95% C.I. = 5.25, 45.22; Figure 8i). No significant difference was observed in the frequency of transitions between any of the WT or Ztm groups (Figure 8j).

Overall, a significant difference in latency to enter the light compartment was observed within the WT female and Ztm female and male BC groups, and for frequency of transitions within the WT and Ztm female BC groups.

## 4. Discussion

The zonulin-expressing mouse model is a relevant model for studying the crosstalk between the microbiome, the intestinal tract, and the brain in the context of neurobehavioral and neuroinflammatory disorders [27,39]. In the present study, we investigated the treatment effects of BC in WT and Ztm mice (Ztm mice characterized by increased intestinal permeability and mild hyperactivity) and compared with WT and Ztm control mice.

Our aim was to study the effect of BC by analyzing the intestinal microbiota and behavior in both colonies. We hypothesized that pursuant to the BC application, intestinal microbial eubiosis could be increased in the Zonulin-expressing mice, which synergize with a dysbiotic, pro-inflammatory microbial community, and, in turn, ameliorate behavioral changes previously reported in Ztm mice [39].

Miranda-Ribera et al. [27,39] previously reported that Ztm has an intrinsic increased intestinal permeability in the small intestine, associated with altered blood–brain barrier integrity at baseline, that influences the development of the immune system. Due to dysbiotic microbial community composition and increased trafficking of pro-inflammatory bacterial species and their products into the systemic circulation, the Ztm mice (female and male) are, therefore, predisposed to inflammation, including neuroinflammation. Moreover, antibiotic depletion of the intestinal microbiota downregulated brain inflammatory markers, ameliorating some anxiety-like behavior in the Ztm mice [27,39].

In the present study, BC administration demonstrated anxiolytic potential and moderately reduced anxiogenic behavior in both WT and Ztm mice. Our findings on latency suggest that BC application reduced aversion to moving from the dark section to the bright-lighted area within the WT female and the Ztm female and male BC groups, and it reduced the frequency of transitions within the Ztm female BC group. The BC treatment may have induced anxiolytic potential, causing reduced latency by increased mobility entering the light compartment, as well as reduced transitions. However, in both cases, we are comparing genotypes with different baselines, except for the significantly reduced latency between the WT female BC and WT female control groups.

These findings correlate with significant differences in the increased abundance and taxonomic diversity of the intestinal microbial community in the WT female mice and taxonomic richness in the Ztm groups (females and males) exposed to BC, compared with the control groups.

### 4.1. Microbiota, Short Chain Fatty Acids, and Behavior in Wild-Type (WT) Mice

The alpha and beta diversity were significantly increased in the WT female mice that underwent BC treatment. Interestingly, many of the increased members of the microbiota produce short-chain fatty acids (SCFAs) in the colon.

At the phylum level, we observed a shift in the Firmicute Bacteroidetes ratio (F/B ratio) within all the BC treatment groups, with an increased F/B ratio in the WT BC females. Among Firmicutes, the *Lachnospiraceae, Lactobacillaceae*, and *Ruminococcaceae* species produce butyrate and other SCFAs. The three major SCFAs produced by intestinal bacteria are butyrate, propionate, and acetate, which can exert several beneficial effects on human metabolism and cognitive function [137,138,139,140,141,142]. Moreover, SCFAs may positively affect intestinal barrier integrity and the host’s immunity [143,144]. Butyrate is produced by various genera (e.g., *Blautia*, *Lachnospira*, *Roseburia*) belonging to the Lachnospiraceae family, and taxa of this family have repeatedly shown their ability to produce beneficial metabolites for the host [138,145]. Butyrate is absorbed/metabolized by the epithelium and may stabilize intestinal barrier protection [145]. Akkermansiaeae, which belongs to the phylum Verrucomicrobia, known to promote intestinal health, and produces both propionate and acetate [142], was found to be increased in the WT female BC group.

Accumulating evidence suggests that these SCFAs can enter the CNS, providing neuroactive properties, though the mechanisms involved in the action of SCFAs on the CNS remain largely unknown [146]. Pre-clinical studies demonstrate that SCFAs exert widespread influence on key neurological and behavioral processes and may be involved in critical phases of neurodevelopmental and neurodegenerative disorders [147,148,149,150,151]. Moreover, SCFAs might directly influence neural function by reinforcing blood–brain barrier integrity, modulating neurotransmission, influencing levels of neurotrophic factors, and promoting memory consolidation. Increased evidence suggests a potential key role of SCFAs in gut–brain axis signaling [146]. However, translating these promising pre-clinical benefits to human neurodevelopmental disorders is challenging [152].

### 4.2. Dysbiosis, Intestinal Barrier, and Behavior in Zonulin Transgenic Mice (Ztm)

Altered microbial composition generally involves decreased abundance and diversity of species and their metabolites, breakdown of the intestinal barrier integrity, and loss of goblet cells. This may result in reduced mucus secretion, thinning of the mucus layer, and, in turn, translocation of pathobionts and toxic metabolites into the blood circulation, which can lead to local and systemic inflammatory responses [23,25,31].

As indicated in the Shannon index, there was a significant difference in microbial community abundances between the Ztm treatment and control groups. PERMANOVA demonstrated significant differences between all groups and within the groups. We observed a decrease in the F/B ratio in the Ztm female BC group compared with the Ztm female control group. Studies reveal that members of the Bacteroidetes and Firmicutes occupy different functional niches in the gut ecosystem, and, therefore, differences between individuals in their relative proportion may lead to large differences in function with relevance for host health [153,154]. The members of the phylum Bacteroidota play an essential role in maintaining the integrity of the interbacterial bonds in the intestines, produce SCFAs (butyrate, propionate, acitate), and are involved in the metabolism of bile acids and the transformation of toxic compounds [142,153,154,155,156]. The Prevotellaceae family, belonging to the Bacteroidota phylum, was increased in the Ztm BC groups. The presence of *Prevotella* in the intestinal microbiota in humans has been inversely correlated with Parkinson’s disease [157,158,159,160]. However, *Prevotella* is a gram-negative bacterium and, therefore, contains lipopolysaccharide (LPS) known to be able to impact human health, primarily through interactions with the immune system, by inducing an innate immune response, specifically through Toll-like receptors (TLRs) [161,162].

Deferribacterota and Desulfobacterota phyla were only found in the Ztm groups, and both BC groups harbored a significantly reduced relative abundance, i.e., Ztm female and male groups. Deferribacterota and Desulfobacterota phyla are involved in the activation of systemic inflammation in the host organism and can cause inflammatory damage and exacerbate energy metabolism disorders [137,163].

### 4.3. Dysbiosis, Behavior, Mental Disorders, and Neuroinflammation

Animal and human studies have demonstrated a clear correlation between altered microbial composition and altered behavior [164,165,166,167,168,169,170,171,172], mental disorders [173,174,175], and the development of neuroinflammation [158,176,177,178,179,180,181]. The intestinal microbiota communicate with the brain via the neural, immune, and metabolic pathways and impact neuronal plasticity and cognition [66,182,183]. This communication takes place either directly via the vagus nerve or indirectly via microbial-derived metabolites, as well as intestinal-derived metabolites, intestinal-derived hormones, and endocrine peptides [3,184].

Human studies have shown that altered microbial composition may affect neurochemical signaling and, therefore, initiate the cascade of pro-inflammatory pathways, which have been linked with depressive outcome [185,186]. Associations between abnormal intestinal microbial commensal compositions and anxiety disorders are well established [41,187]. Several reports show that experimental manipulations altering the intestinal microbiota impact anxiety-like behavior related to inflammatory status [177,188,189,190,191,192,193,194,195,196,197].

Chen et al. showed that Lachnospiraceae and Ruminococcaceae were less abundant in subjects diagnosed with anxiety disorders compared with healthy controls [41]. Both taxa are butyrate producers and may enhance the intestinal barrier by suppressing inflammation [198]. In the present study, the taxa, as mentioned above, were significantly increased in both WT female and male BC groups, and *Ruminococcus* was increased in both Ztm female and male BC groups. 

Alzheimer’s disease (AD) has been associated with a reduction in Bacteroidetes and an increase in the F/B ratio [199,200], and a higher level of Firmicutes has been reported in patients with mild cognitive impairment (MCI) [201]. Inconsistent with this evidence, a decrease in Firmicutes in AD [202,203] has also been reported. Leu and coworkers [202] reported an increased abundance of Bacteroidetes in amnestic MCI, yet no difference was recorded when comparing AD patients and healthy controls. These inconsistent findings might reflect the dynamic changes of the intestinal microbiota during each stage of cognitive dysfunction.

An increase in some bacteria belonging to the phylum Firmicutes, including Ruminococcaceae and Enterococcaceae, has been correlated with neuroinflammation [200,201]. However, there is evidence of an abundance of Clostridiaceae, Ruminococcaceae, Eubacteriaceae, and Veillonellaceae in subjects absented from neuroinflammation, in conjunction with normal cognitive function [200,202,204]. An increase in Enterobacteriaceae, belonging to the Proteobacteria phylum, has been shown to correlate with cognitive impairment in several studies [201,202,204]. However, since many classes of bacteria with contrary characteristics are subordinated to one phylum, unambiguous results on the phylum level are scarce.

Findings from our study reveal that BC treatment modulates the intestinal microbial communities in WT mice towards a preponderance of potentially beneficial species. These microbial shifts possibly prevented anxiogenic behavior in the WT female mice. According to our data, the WT female BC group harbored a significantly increased abundance of the anti-inflammatory bacteria *Eubacterium* [205], an increased abundance of the anti-inflammatory bacteria *Ruminococcus* (research has shown reduced abundance in major depressive disorder (MDD) [206,207,208,209] and in Parkinson’s Disease (PD) [210])), increased abundance of the *Eubacterium Xylanophilum* group (reduced abundance has been found in ASD [211,212]), and significantly reduced abundance of the pro-inflammatory bacteria *Tyzzerella* [213,214], when compared with the WT female control group. Animal and human research show that the relative abundances of the genera *Tyzzerella* correlates with circulating pro-inflammatory cytokines (IL-1β) and certain behavioral outcomes, such as lethargy and anxiety-like behavior in chemotherapy-induced inflammation in female mice [213], as well as increased abundance by eightfold in neurogenerative diseases, such as in pediatric multiple sclerosis in humans [214].

Moreover, the BC treatment shifted the dysbiotic microbial community towards eubiosis in the Ztm mice, i.e., increased the balance within the intestinal microbial ecosystem by reducing the abundance of potentially pathogenic species. The increased eubiosis in the Ztm female BC group was paralleled by the BC anxiolytic effects. The group harbored a significantly reduced abundance of the pro-inflammatory genera *Peptococcus* [215,216,217] and Bifidobacterium (research shows an increased abundance of Bifidobacteria in ADHD [55,218,219]; in MDD [220,221,222]; and in bipolar (BP) [220,223]; in Schizophrenia [224,225]; yet reduced abundance in AD [52,211,226,227,228,229])). The Ztm female BC group harbored a significantly reduced abundance of *Desulfovibrio* (an increased abundance is known in BP [220,223], and in ASD [52,211,229,230])*,* reduced abundance of *Blautia* (reduced abundance known in PD [210], yet an increased abundance in MDD [205,209,231]) when compared with the Ztm female control group.

The increased abundance of the eubiotic intestinal microbial community within the Ztm male BC group was likewise paralleled by moderate anxiolytic effects. The BC group harbored a significantly reduced abundance of *Desulfovibrio* (associated with the severity of symptoms in ASD [232]) and increased abundance of *Ruminococcus*, an important contributor to the intestinal ecosystem [233]. An eubiotic microbiome [234] has been shown to decrease intestinal hyperpermeability and mucosal inflammatory markers in both mice [235] and humans [236]. Relieving dysbiosis and eliminating pathobionts by increasing intestinal eubiosis can elevate the production of favorable metabolites, mucus production, and protection against inflammation levels [237].

### 4.4. Bovine Colostrum, Oligosaccharides, Short-Chain FattyAacids (SCFAs), and Eubiosis

BC has been researched for its main constituents [87] in gastrointestinal health and disease [99], for supporting immune and digestive health [105], its effects on enteric bacteria [238], and in pediatrics [102]. Moreover, BC is rich in prebiotic components, such as oligosaccharides and glycans [87,89,239]. Prebiotics are substrates that are selectively utilized by the host’s eubiotic commensal bacteria and confer a health benefit [240]. Oligosaccharides and glycans provide the host with favorable metabolites, such as SCFAs [241,242,243,244]. 

Oligosaccharides have been demonstrated to reduce stress responsiveness and anxiety and facilitate changes in hippocampal synaptic efficacy [245,246,247,248]^.^ Research has shown that oligosaccharides can significantly increase *Lactobacillus, Bacteroides,* and *Bifidobacterium* spp. [249,250,251]. Moreover, research shows the above taxa can reduce the production of proinflammatory cytokines and increase anti-inflammatory cytokines, which may attenuate post-inflammatory anxiety [3]. Oligosaccharides are known to prevent an LPS-mediated increase in cortical 5-HT2A receptor and IL1-β levels in mice [3]. Administration of oligosaccharides has been known to induce suppression of the neuroendocrine stress response and to increase the processing of positive versus negative attentional vigilance, thus resulting in an early anxiolytic-like phenotype [3].

*γ*-Aminobutyric acid (GABA) is a major inhibitory neurotransmitter of the vertebrate central nervous system [252] and the main inhibitory neurotransmitter in the brain [253]. Dysfunctions in the GABA system are implicated in anxiety disorders [252]. According to Berding et al. [65], dietary intervention applying prebiotic foods improved perceived stress in a healthy population, while eliciting specific metabolic changes in the intestinal microbiota. A transcriptome analysis of human fecal samples from healthy individuals showed that GABA-producing pathways are actively expressed by Bacteroides [254], and research shows that certain strains of Lactobacillus secrete GABA [255]. The extent to which pre- and probiotic [256] foods might be therapeutically useful in patients with clinically recognized anxiety disorders is presently unknown, which provides a reasonable rationale for exploring their potential value further [257].

In a randomized, double-blind, placebo-controlled trial, the effect of early enteral BC supplementation on intestinal permeability in critically ill patients was assessed. Plasma endotoxin concentration decreased significantly in the treatment group (*p* < 0.05), and zonulin plasma levels reduced significantly compared with the placebo group (*p* < 0.001) [109]. In another systematic review, the BC supplementation effect on increased intestinal permeability in athletes was assessed and showed beneficial effects for reverting intestinal hyperpermeability [258]. 

In a small pilot study, a probiotic/colostrum supplementation was tested on intestinal function in children diagnosed with ASD [259]. All study participants experienced a reduction in at least one gastrointestinal symptom in at least one treatment arm of the study, and some of the participants reported a reduced frequency of specific gastrointestinal symptoms, as well as of the occurrence of aberrant behaviors. However, no treatment effect on any specific microbial genera was found, as the participants’ microbiota baseline did not shift much throughout the study period. No significant changes were observed for any of the urine or serum metabolites.

In conclusion, further investigation is needed to examine the interdependencies among the components of MGBA and the potential value of BC for human use. The main limitations of this study were the small sample size in each group and limited analysis of the raw material, such as basic chemical composition and no microbiological analysis performed.

### 4.5. Future Perspectives

Studies showing the prebiotic influence on brain physiology and behavior are primarily descriptive. Further studies are warranted to understand mechanisms. The focus should be on which commensal microbial-derived metabolites are involved, as well as on the pathways by which these effects can be mediated. Moreover, chemical and microbiological analyses are of importance, as differences in the BC collection period may cause bioactive variability, as well as variation in processing, pasteurization, and storage conditions. When researching BC as a therapeutic agent for a medical condition, consistency of the product is vital. Based on our findings and the considerations outlined above, original research on BC application is needed from well-conducted, clinical, randomized control trials on human/clinical use in order to evaluate long-term safety/efficacy and to determine the optimal dose. Benefits and challenges need to be addressed to fully understand the potential value of BC for both prophylactic and clinical use.

## 5. Conclusions

In conclusion, we found significant differences in the microbiota in all BC groups compared with control groups. Moreover, our findings reveal that BC treatment modulates the intestinal microbial communities in WT and Ztm mice towards a preponderance of potentially beneficial species, and it shifts the dysbiotic microbial community towards eubiosis by reducing the abundance of potentially pathogenic species. The increased microbial ecosystem balance seemed to create a mild anxiolytic effect, and it also possibly prevented anxiogenic effect in the WT female group and reduced anxiogenic behavior in Ztm female and male mice. Colostrum application seems to support a healthy F/B ratio and significantly increase the abundance of anti-inflammatory microbial commensal bacteria, such as Lachnospiraceae, Prevotellaceae, Ruminococcaceae, Akkermansiaceae, the *Eubacterium* xylanophilum group, and *Lactococcus*. Moreover, BC application seems to significantly reduce various pro-inflammatory bacterial species, such as *Deferribacterota*, *Desulfobacterota*, *Tyzzerella, Peptococcus,* and *Enterococcaceae*. Regarding the evidence of MGBA interactions, BC treatment could be considered for prophylactic approaches in the future. However, further research is needed to explore the interdependencies among the intestinal microbiota, eubiosis, pathobionts, and neuroinflammation, as well as the potential value of BC for human use.

## Figures and Tables

**Figure 1 biomedicines-11-00091-f001:**
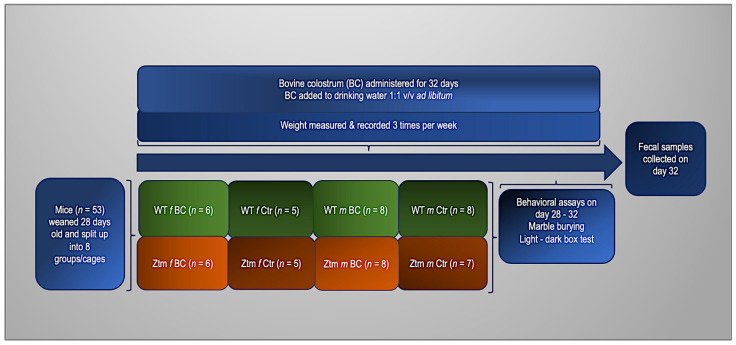
Experimental timeline and flowchart. WT and Ztm mice in separate cages, females and males, receiving either BC added to drinking water (1:1 *v*/*v*) or water only ad libitum for 32 days. Behavioral assays performed on day 28–32 and fecal samples collected on day 32. BC = bovine colostrum treatment; WT = wild-type mice; Ztm = zonulin transgenic mice; Ctr = control mice, *f* = female; *m* = male; *v*/*v* = volume per volume, ad libidum = as much or as often as necessary or desired; *n* = number of mice in each group.

**Figure 2 biomedicines-11-00091-f002:**
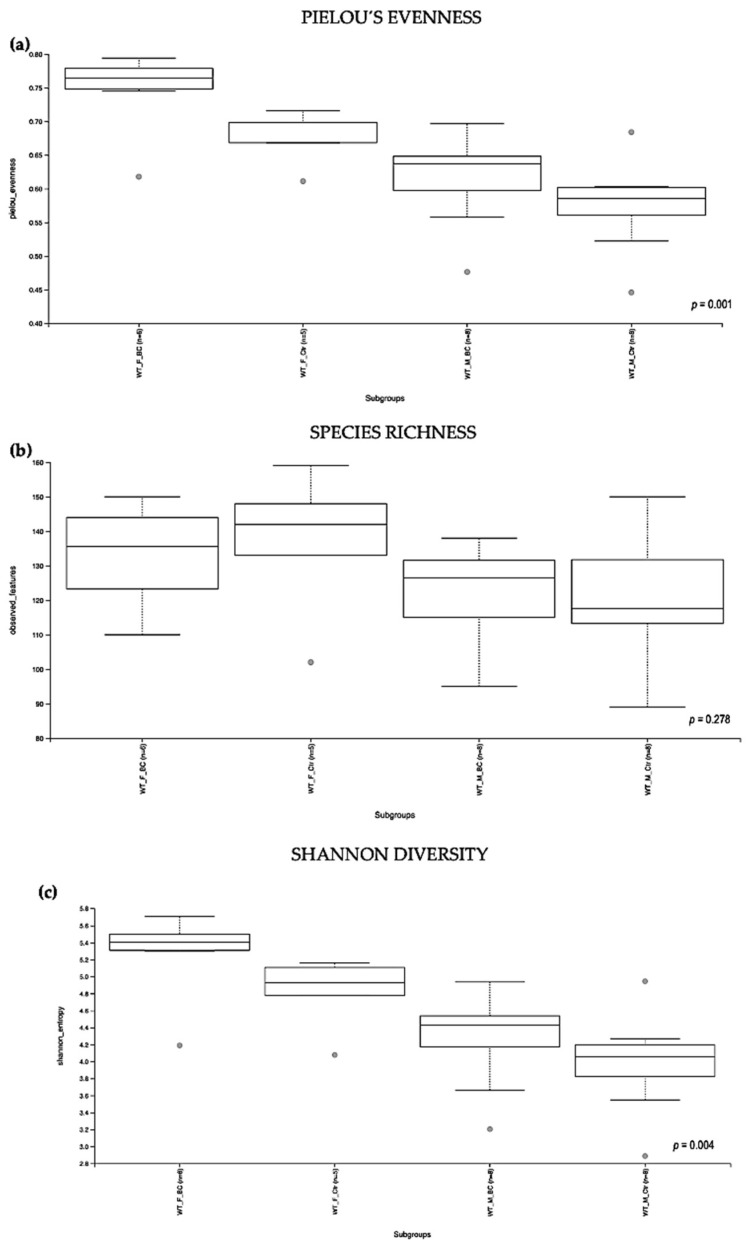
Alpha diversity of the intestinal microbial communities between WT BC and control groups post BC treatment. Comparison of boxplots depicting Pileou´s evenness (**a**), species richness (**b**), and Shannon diversity (**c**). Diversity among WT *f* BC (*n* = 6), WT *f* Ctr (*n* = 5), WT *m* BC (*n* = 8), and WT *m* Ctr (*n* = 8) groups. *p* < 0.05 indicating that the alpha diversity between groups is significant. *p* values = Kruskal–Wallis rank-sum test (all groups). WT = wild-type mice; BC = bovine colostrum treatment; Ctr = control mice; *f* = female; *m* = male; *n* = number of mice in a group.

**Figure 3 biomedicines-11-00091-f003:**
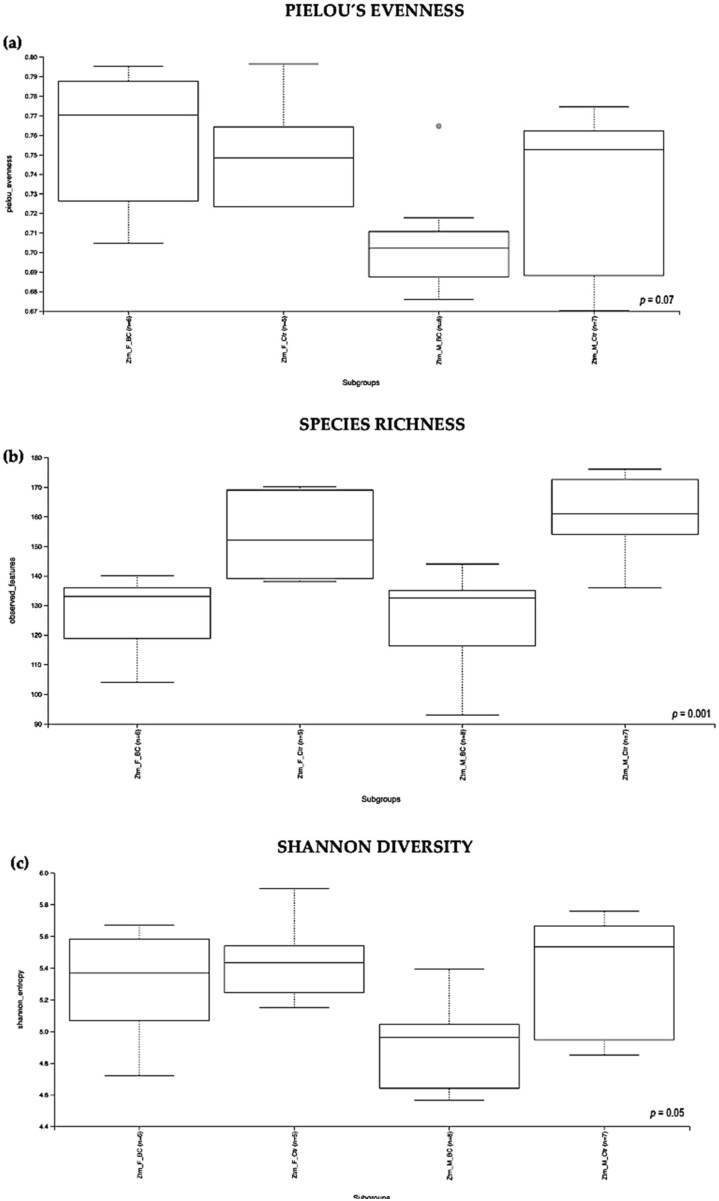
Alpha diversity of the intestinal microbial communities between Ztm BC and control groups post BC treatment. Comparison of boxplots depicting Pileou´s evenness (**a**), species richness (**b**), and Shannon diversity (**c**). Diversity among Ztm *f* BC (*n* = 6), Ztm *f* Ctr (*n* = 5), Ztm *m* BC (*n* = 8), and Ztm *m* Ctr (*n* = 7) groups. *p* < 0.05 indicating that the alpha diversity between groups is significant. *p* values = Kruskal–Wallis rank-sum test (all groups). WT = wild-type mice; BC = bovine colostrum treatment; Ctr = control mice; *f* = female; *m* = male; *n* = number of mice in a group.

**Figure 4 biomedicines-11-00091-f004:**
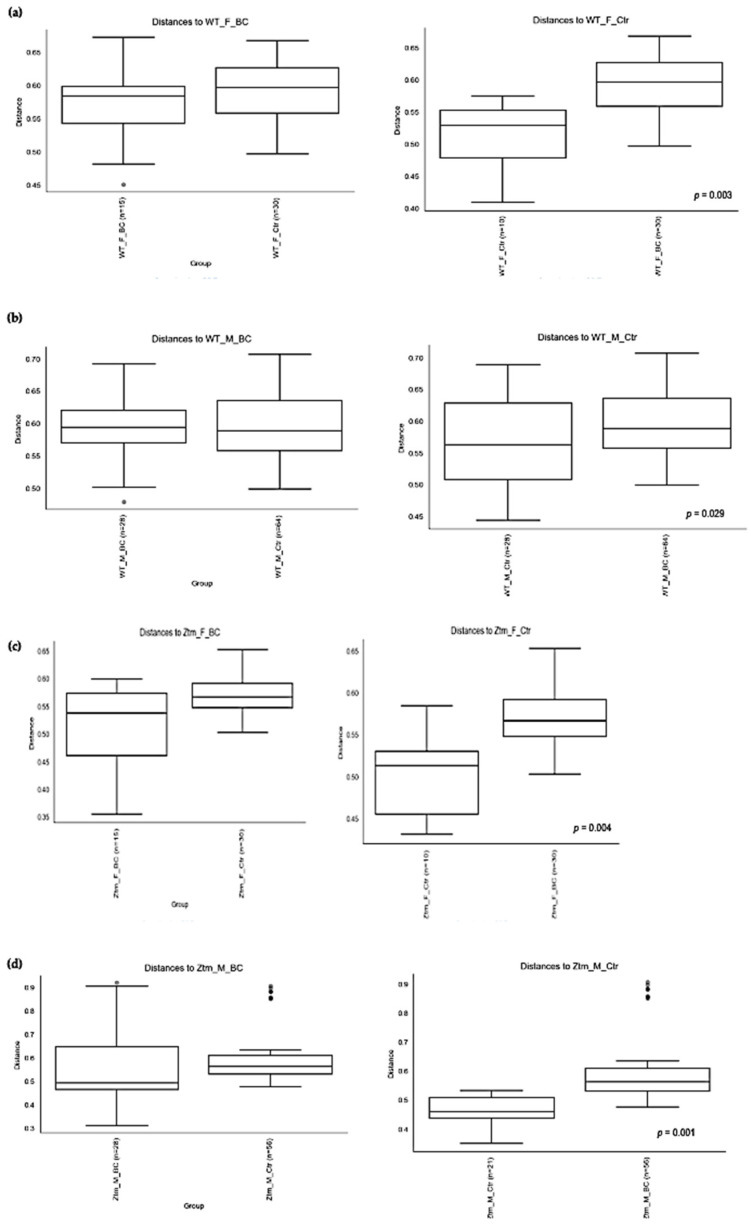
Beta diversity of the intestinal microbial communities between WT f BC and WT f control groups (**a**), WT m BC and WT m control groups (**b**), Ztm f BC and Ztm f control groups (**c**), and Ztm m BC and Ztm m control groups. (**d**) Box plot showing Jaccard distance between groups and within groups, *p* < 0.05 indicating beta diversity between groups is significant. BC = bovine colostrum treatment; Ctr = control mice; WT = wild-type mice; Ztm = zonulin transgenic mice; *f* = female; *m* = male; *n* = number of mice in a group. *p* = pairwise PERMANOVA results.

**Figure 5 biomedicines-11-00091-f005:**
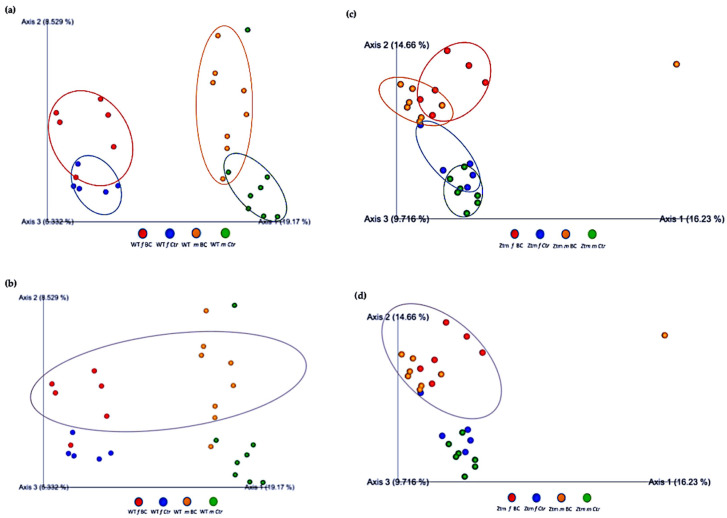
Beta diversity of intestinal microbial communities between WT BC and control groups, and Ztm BC and control groups. PCA plot based on Jaccard distance. Each dot on the plot represents one sample from each group. The Jaccard patterns indicate BC and Ctr representing; (**a**): WT BC (*f* & *m*) vs. WT Ctr (*f* & *m*) partially overlapping; (**b**): WT BC groups (*f* & *m*); (**c**): Ztm BC (*f* & *m*) vs. Ztm Ctr (*f* & *m*) distinct clustering; (**d**): Ztm BC groups (*f* & *m*). The distance between dots indicates the degree of similarity of taxonomic composition of samples. BC = bovine colostrum treatment; Ctr = Control mice; WT = wild-type mice; Ztm = zonulin transgenic mice; *f* = female; *m* = male; PCA = principal component analysis.

**Figure 6 biomedicines-11-00091-f006:**
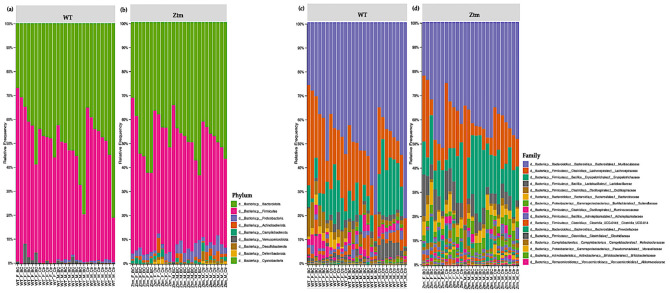
Taxonomy representation at phyla (**a**), (**b**) and family (**c**), (**d**) level. The abscissa represents the groups, and the ordinate represents the relative abundance of intestinal bacteria in WT and Ztm mice (*f* & *m*). (**a**): Relative percentage of most abundant phyla in each sample between WT BC groups (*n* = 14) and WT control groups (*n* = 13). (**b**): Relative percentage of most abundant phyla in each sample between Ztm BC groups (*n* = 14) and Ztm control groups (*n* = 12). (**c**): Relative abundance of bacteria at family level in WT BC and control groups. (**d**): Relative abundance of bacteria at the family level in Ztm BC and control groups. *n* = 53. WT = wild-type mice; Ztm = zonulin transgenic mice; BC = bovine colostrum treatment; Ctr = control mice; F = female; M = male.

**Figure 7 biomedicines-11-00091-f007:**
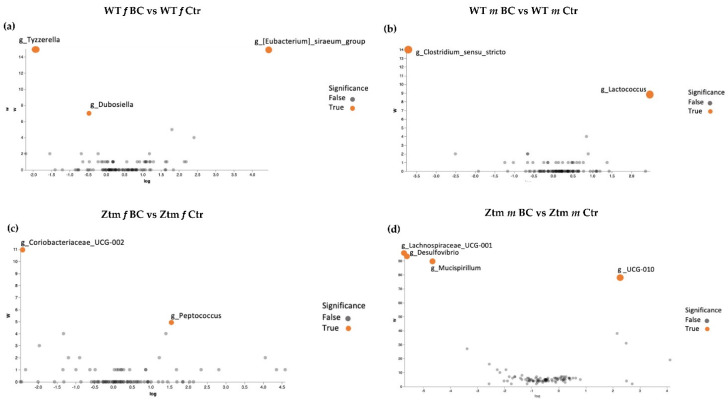
ANCOM volcano plot of statistical differences between treatment groups (WT BC & Ztm BC) and control groups (WT Ctr & Ztm Ctr) at genus level. (**a**): WT *f* BC harbor a significant increased abundance of *Eubacterium* (Ruminococcaceae family) and a significantly reduced abundance of *Tyzzerella* (Lachnospiraceae family) and *Dubosiella* (Erysipelotrichaceae family) compared with control group. (**b**): WT *m* BC harbor a significantly increased abundance of *Lactococcus* (Streptococcaceae family) and a significantly reduced abundance of *Clostridium sensu stricto* 1 (Clostridiaceae family) compared with control group. (**c**): Ztm *f* BC harbor a significantly reduced abundance of *Coriobacteriaceae* UCG-002 (Atopobiaceae family) and *Peptococcus* (Peptococcaceae family) compared with control group. (**d**): Ztm *m* BC harbor a significantly reduced abundance of *Lachnospiraceae* UCG-001 (order Clostridia), *Desulfovibrio* (Desulfovibrionacea family), and *Mucispirillum* (Deferribacteraceae family) and a significant increased abundance of UCG-010 (order Oscillospirales) compared with control group. WT = wild-type mice; Ztm = zonulin transgenic mice; Ctr = control mice; BC = bovine colostrum treatment; *f* = female; *m* = male.

**Figure 8 biomedicines-11-00091-f008:**
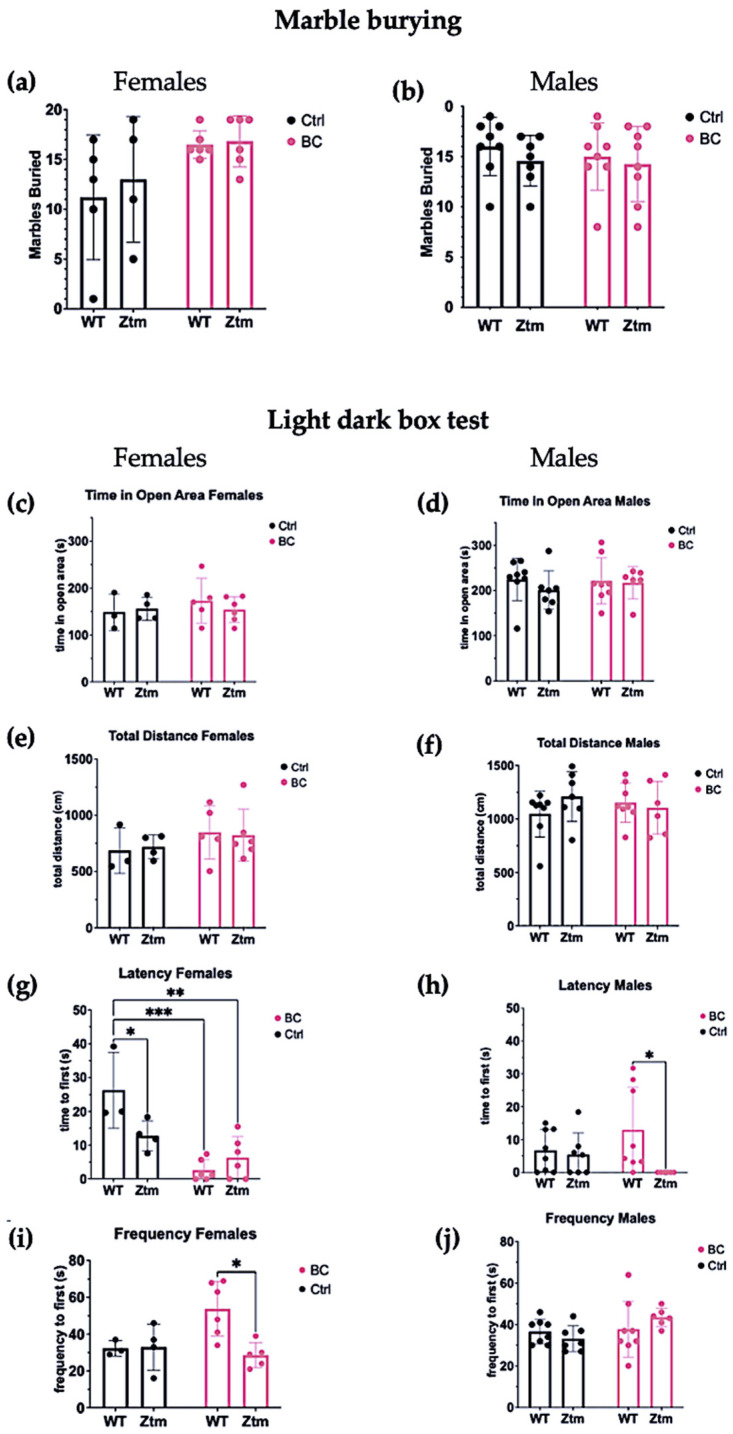
A behavior profile of WT and Ztm mice measuring treatment effects post BC application. Marble burying (**a**,**b**): No significant difference was observed in total number of marbles buried between the groups. Light dark box (**c**–**j**): No significant difference was observed for time spent in open area between any of the groups (**c**,**d**). No significant difference was observed in total distance traveled between the groups (**e**,**f**).Within the female groups, latency to enter the light section was significantly different between WT BC and WT control groups (*p* = 0.0003), and between Ztm BC and WT control groups (*p* = 0.001) (**g**). A significant difference was observed in latency to enter the light section between WT male BC and Ztm male BC groups (*p* = 0.03) (**h**). There was a significant difference in frequency of transitions between WT female BC and Ztm BC groups (*p* = 0.01) (**i**). No significant difference was observed in frequency of transitions between any of the WT or Ztm male groups (**j**). *n* = 53. WT = wild-type mice; BC = bovine colostrum treatment; Ctrl = control mice; Ztm = zonulin transgenic mice; *n* = total number of mice. * *p* < 0.05; ** *p* < 0.005; *** *p* < 0.0005; *p* values adjusted with Tukey´s HSD.

**Table 1 biomedicines-11-00091-t001:** Alpha diversity of the intestinal microbiota in WT mice post BC treatment. Quantified using Pielou´s evenness, species richness, and Shannon diversity indexes, which relate OTU richness and evenness, the total number of observed species, and variation and complexity within the group. Alpha diversity analysis performed with Kruskal–Wallis rank-sum and pairwise tests. WT = wild- type mice; BC = bovine colostrum treatment; Ctr = control mice; *f* = female; *m* = male; *n* = number of mice in each group.

Pielou’s Evenness				
Group 1	Group 2	H	*p*-value	q-value
WT BC (*n* = 14) *	WT Ctr (*n* = 13)	15.549	0.001	-
WT *f* BC (*n* = 6) **	WT *f* Ctr (*n* = 5)	4.033	0.044	0.053
	WT *m* BC (*n* = 8)	6.016	0.014	** 0.028 **
	WT *m* Ctr (*n* = 8)	8.816	0.002	** 0.017 **
WT *f* Ctr (*n* = 5) **	WT *m* BC (*n* = 8)	4.200	0.040	0.053
	WT *m* Ctr (*n* = 8)	6.942	0.008	** 0.025 **
WT *m* BC (*n* = 8) **	WT *m* Ctr (*n* = 8)	2.161	0.141	0.141
**Species Richness**				
** Group 1 **	** Group 2 **	** H **	***p*-value**	**q-value**
WT BC (*n* = 14) *	WT Ctr (*n* = 13)	3.849	0.278	-
WT *f* BC (*n* = 6) **	WT *f* Ctr (*n* = 5)	0.209	0.647	0.751
	WT *m* BC (*n* = 8)	1.845	0.174	0.374
	WT *m* Ctr (*n* = 8)	1.209	0.271	0.407
WT *f* Ctr (*n* = 5) **	WT *m* BC (*n* = 8)	2.600	0.106	0.374
	WT *m* Ctr (*n* = 8)	1.740	0.187	0.374
WT *m* BC (*n* = 8) **	WT *m* Ctr (*n* = 8)	0.100	0.751	0.751
**Shannon Diversity**				
** Group 1 **	** Group 2 **	** H **	***p*-value**	**q-value**
WT BC (*n* = 14) *	WT Ctr (*n* = 13)	12.932	0.004	-
WT *f* BC (*n* = 6) **	WT *f* Ctr (*n* = 5)	4.033	0.044	0.066
	WT *m* BC (*n* = 8)	5.400	0.020	0.060
	WT *m* Ctr (*n* = 8)	8.066	0.004	** 0.027 **
WT *f* Ctr (*n* = 5) **	WT *m* BC (*n* = 8)	3.085	0.078	0.094
	WT *m* Ctr (*n* = 8)	4.200	0.040	0.066
WT *m* BC (*n* = 8) **	WT *m* Ctr (*n* = 8)	1.863	0.172	0.172

* Rank-Sum test; ** Pairwise test.

**Table 2 biomedicines-11-00091-t002:** Alpha diversity of the intestinal microbiota in Ztm mice post BC treatment. Quantified using Pielou´s evenness, species richness, and Shannon diversity indexes, which relate OTU richness and evenness, the total number of observed species, and variation and complexity within the group. Alpha diversity analysis were performed with Kruskal–Wallis rank-sum and pairwise tests. Ztm = zonulin transgenic mice; BC = bovine colostrum treatment; Ctr = control mice; *f* = female; *m* = male; *n* = number of mice in each group.

Pielou’s Evenness				
Group 1	Group 2	H	*p*-value	q-value
Ztm BC (*n* = 14) *	Ztm Ctr (*n* = 12)	6.960	0.073	-
Ztm *f* BC (*n* = 6) **	Ztm *f* Ctr (*n* = 5)	0.000	1.000	1.000
	Ztm *m* BC (*n* = 8)	4.816	0.028	0.084
	Ztm *m* Ctr (*n* = 7)	2.469	0.116	0.232
Ztm *f* Ctr (*n* = 5) **	Ztm *m* BC (*n* = 8)	5.485	0.019	0.084
	Ztm *m* Ctr (*n* = 7)	0.323	0.569	0.683
Ztm *m* BC (*n* = 8) **	Ztm *m* Ctr (*n* = 7)	0.482	0.487	0.683
**Species Richness**				
** Group 1 **	** Group 2 **	** H **	***p*-value**	**q-value**
Ztm BC (*n* = 14) *	Ztm Ctr (*n* = 12)	15.359	0.001	-
Ztm *f* BC (*n* = 6) **	Ztm *f* Ctr (*n* = 5)	5.659	0.017	** 0.026 **
	Ztm *m* BC (*n* = 8)	0.016	0.897	0.897
	Ztm *m* Ctr (*n* = 7)	7.449	0.006	** 0.019 **
Ztm *f* Ctr (*n* = 5) **	Ztm *m* BC (*n* = 8)	6.580	0.010	** 0.020 **
	Ztm *m* Ctr (*n* = 7)	0.797	0.371	0.446
Ztm *m* BC (*n* = 8) **	Ztm *m* Ctr (*n* = 7)	9.053	0.002	** 0.015 **
**Shannon Diversity**				
** Group 1 **	** Group 2 **	** H **	***p*-value**	**q-value**
Ztm BC (*n* = 14) *	Ztm Ctr (*n* = 12)	7.609	0.054	-
Ztm *f* BC (*n* = 6) **	Ztm *f* Ctr (*n* = 5)	0.300	0.583	0.775
	Ztm *m* BC (*n* = 8)	3.750	0.052	0.158
	Ztm *m* Ctr (*n* = 7)	0.081	0.775	0.775
Ztm *f* Ctr (*n* = 5) **	Ztm *m* BC (*n* = 8)	6.942	0.008	0.050
	Ztm *m* Ctr (*n* = 7)	0.164	0.684	0.775
Ztm *m* BC (*n* = 8) **	Ztm *m* Ctr (*n* = 7)	3.013	0.082	0.165

* Rank-Sum test; ** Pairwise test.

**Table 3 biomedicines-11-00091-t003:** Beta diversity of the intestinal microbiota. Beta diversity showing the degree of pair-wise similarity in species composition between the groups assessed by Bray–Curtis and Jaccard matrices. Similarity measures of abundance, presence, and absence data of species, genera, families, orders, classes, and phyla was assessed in the four groups, WT (BC & Ctr) and Ztm (BC & Ctr), post BC treatement. WT = wild-type mice; BC = bovine colostrum treatment; Ctr = control mice; *f* = female; *m* = male; *n* = number of mice in each group.

Beta Diverstiy Index	Permanova	Group 1	Group 2	*n*	Permutations	Pseudo-F	*p*-Value
**Bray Curtis**	**Groups**	WT BC	WT Ctr	27	999	2.428	**0.034**
		Ztm BC	Ztm Ctr	26	999	4.753	**0.001**
	**Subgroups**	WT *f* BC	WT *f* Ctr	11	999	2.079	0.075
		WT *m* BC	WT *m* Ctr	16	999	2.653	**0.033**
		Ztm *f* BC	Ztm *f* Ctr	11	999	4.137	**0.012**
		Ztm *m* BC	Ztm *m* Ctr	15	999	3.112	**0.001**
**Jaccard**	**Groups**	WT BC	WT Ctr	27	999	1.561	**0.043**
		Ztm BC	Ztm Ctr	26	999	3.455	**0.001**
	**Subgroups**	WT *f* BC	WT *f* Ctr	11	999	1.967	**0.003**
		WT *m* BC	WT *m* Ctr	16	999	1.501	**0.029**
		Ztm *f* BC	Ztm *f* Ctr	11	999	2.323	**0.004**
		Ztm *m* BC	Ztm *m* Ctr	15	999	2.895	**0.001**

## Data Availability

Not applicable. All data generated or analyzed during this study are included in this published article.

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
