# Peer review of "Prophylactic Effect of Bovine Colostrum on Intestinal Microbiota and Behavior in Wild-Type and Zonulin Transgenic Mice"

_biomedicines, 2022, doi:10.3390/biomedicines11010091_

Round 1

Reviewer 1 Report

Asbjornsdottir et al studied Prophylactic Effect of Bovine Colostrum on behavioural changes and intestinal microbiota in Zonulin transgenic mice

1. Abstract is not clear and not precise. Results not written clear (line 34 and 35) confusing. Message is not conveyed clearly. 

2. Extensive number of references. Use only appropriate reference and reduce number to 50 or less

3. Manuscript is very length and difficult to read. It should be revised and deliver in precise as well as readable format.

4. Figure qualities are very bad (fig 1and 2) and need to be replaced

5. Ethical clearance was obtained in 2013. Is valid for this all experiment of study. When study was conducted? Author should confirm the same.

Author Response

Reviewer #1

Extensive editing of English language and style required.

Thank you for your suggestion, this has been taken care of.

Comments and Suggestions for Authors

Asbjornsdottir et al studied Prophylactic Effect of Bovine Colostrum on behavioral changes and intestinal microbiota in Zonulin transgenic mice.

  1. Abstract is not clear and not precise. Results not written clear (line 34 and 35) confusing. Message is not conveyed clearly. 

Thank you for your comment, we have added BC for WT male mice in line 36 to make this more accurate.

  1. Extensive number of references. Use only appropriate reference and reduce number to 50 or less.

Thank you for your comment. However, the authors agree on the importance and relevance of the references. As the two additional reviewers agreed on the relevance of all cited references and did not ask for any reduction, and the number of references is within the limit of the journal requirements, we would appreciate your reconsideration.

  1. Manuscript is length and difficult to read. It should be revised and deliver in precise as well as readable format. Thank you for your comment, this has been addressed accordingly throughout the manuscript and the main points highlighted where needed.

  1. Figure qualities are bad (fig 1 and 2) and need to be replaced. Thank you, we did submit all figures in the requested resolution and very good quality. However, these will be replaced by the editor when the manuscript has been accepted.

  1. Ethical clearance was obtained in 2013. Is valid for this all experiment of study. When study was conducted? Author should confirm the same.

Thank you for your comment. The permission was obtained for this study in 2020 as is already stated in the manuscript, see section at the end of the manuscript “Institutional Review Board Statement:” The animal study protocol was approved by the Institutional Review Board (Institutional Animal Care and Use Committee) of Massachusetts General Hospital (protocol code 2013N000013; 3 March 2021).

Reviewer 2 Report

The authors evaluated the prophylactic effect of bovine colostrum (BC) on intestinal microbiota and behavior in wild-type and zonulin-transgenic mice.

It is a well-written paper based on a well-designed experimental protocol. Microbiota in all cases was evaluated by sequencing and analyzed by bioinformatics. Benefic bacteria taxonomy abundance and diversity were increased under the consumption of BC in WT mice models . In transgenic dysbiotic mice, the microbial profile changed to eubiosis . Additionally, BC induced an anxiolytic effect in WT female mice and lower anxiogenic behavior in the transgenic female and male mice. The value of BC consumption is underlined in their research. Yet, they discussed the role of the intestinal microbiota in health and disease and its possible connection to neuroinflammation via gut-brain axis communication.

It is a really interesting article dealing with hot questions and is well written, and presented based on an extended bibliography.

The authors have to recheck their bibliography pe. 61 and 72 (there are no author's names),192 (names are in capitals while in the other articles are not).

My suggestion is to ACCEPT and publish the paper after these minor corrections.

Author Response

English language and style are fine/minor spell check required.

Thank you, this has been taken care of.

Comments and Suggestions for Authors

The authors evaluated the prophylactic effect of bovine colostrum (BC) on intestinal microbiota and behavior in wild-type and zonulin-transgenic mice.

It is a well-written paper based on a well-designed experimental protocol. Microbiota in all cases was evaluated by sequencing and analyzed by bioinformatics. Benefic bacteria taxonomy abundance and diversity were increased under the consumption of BC in WT mice models.

In transgenic dysbiotic mice, the microbial profile changed to eubiosis. Additionally, BC induced an anxiolytic effect in WT female mice and lower anxiogenic behavior in the transgenic female and male mice.

The value of BC consumption is underlined in their research. Yet, they discussed the role of the intestinal microbiota in health and disease and its possible connection to neuroinflammation via gut-brain axis communication.

It is a really interesting article dealing with hot questions and is well written and presented based on an extended bibliography.

The authors have to recheck their bibliography pe. 61 and 72 (there are no author's names),192 (names are in capitals while in the other articles are not).

Thank you for your comment, this has been taken care of.

My suggestion is to ACCEPT and publish the paper after these minor corrections.

Thank you for your comments and kind words.

Reviewer 3 Report

This article has scientific quality, and I think that the survey will find an interested audience among the journal readers.

I have the following queries/comments:

L. 59: “….and even the brain”. Please revise it.

L 67-68:  “The release of zonulin from the intestinal mucosa is triggered by several environmental stimuli, including intestinal dysbiosis”. Is intestinal dysbiosis environmental stimuli?

L 74-75: “Defects in the intestinal barrier function, including dysbiosis”. Is dysbiosis  part of the pathophysiology of the intestinal epithelium/wall? Please revise it.

L 89-90: “understanding environmental intestinal microbiota modulation. May is better the phrase active modulation through external exposome.

L 122: previously.[115]. Remove the dot.

L 175: Microbiological analysis of BC?

L 180 “Two standard behavioral assays, MB [120,121] and the LDB [116–119] test  ..” Please write that clarifications are given immediately below.

L 188:  Animals, rodents, mice ….   Please follow a uniform naming.

Give  better resolution for  Figure 2, 3 & 4.

L 409: Bacteroidetes

L 429: Campylobacterota instead of Campylobacteria

L 437:  Don’t use italics at the family level, only from the genus level and below

You only researched the basic chemical composition of colostrum, but what about amino acid composition, fatty acid analysis?

The text corresponding to the discussion section is considered incomplete in relation to the labyrinthine section of the results. Furthermore, you talk about topics that you have not included in your experimental protocol, such as Oligosaccharides, SCFAs.

Author Response

Reviewer #3

I don't feel qualified to judge about the English language and style.

Thank you for your comment.

Comments and Suggestions for Authors

This article has scientific quality, and I think that the survey will find an interested audience among the journal readers.

Thank you for your constructive comment and kind words.

I have the following queries/comments:

  1. 59: “….and even the brain”. Please revise it.

Thank you, we have replaced the word “even” with “as well as”.

L 67-68: “The release of zonulin from the intestinal mucosa is triggered by several environmental stimuli, including intestinal dysbiosis”. Is intestinal dysbiosis environmental stimuli?

Thank you for your comment. Yes, it can be considered as such in the internal/intestinal environment.

L 74-75: “Defects in the intestinal barrier function, including dysbiosis”. Is dysbiosis part of the pathophysiology of the intestinal epithelium/wall? Please revise it.

Thank you for your comment, we have added the following text for clarification in line 70; “The mucosal barrier is of physical, biochemical, and immune nature and the microbiota can be considered as part of this system because of the mutual influence occurring between the host and the luminal microorganisms.”

L 89-90: “understanding environmental intestinal microbiota modulation”. May is better the phrase active modulation through external exposome.

Thank you for this valuable comment, we have added this to our text.

L 122: previously.[115]. Remove the dot. 

Thank you, this has been fixed.

L 175: Microbiological analysis of BC? 

Thank you for your question, this was not analyzed.

L 180 “Two standard behavioral assays, MB [120,121] and the LDB [116–119] test ..” Please write that clarifications are given immediately below.

Thank you, this has been added.

L 188:  Animals, rodents, mice ….   Please follow a uniform naming.

Thank you, we have replaced rodents with mice.

Give better resolution for Figure 2, 3 & 4. 

Thank you, we did submit all figures in the requested resolution and very good quality. However, these will be replaced by the editor when the manuscript has been accepted.

L 409: Bacteroidetes.

Thank you, this has been fixed.

L 429: Campylobacterota instead of Campylobacteria.

Thank you, this has been fixed.

L 437:  Don’t use italics at the family level, only from the genus level and below.

Thank you, this has been fixed.

You only researched the basic chemical composition of colostrum, but what about amino acid composition, fatty acid analysis?

Thank you, there were no further analysis performed.

The text corresponding to the discussion section is considered incomplete in relation to the labyrinthine section of the results. Furthermore, you talk about topics that you have not included in your experimental protocol, such as Oligosaccharides, SCFAs.

Thank you for your comment. The reason for including oligosaccharide and SCFAs in the discussions is due to the unforeseen effects of BC on the microbiome, i.e., factors in the BC such as oligosaccharides which may stimulate the production of SCFAs which can in turn affect behavior.

Round 2

Reviewer 1 Report

I have no further queries. I would suggest authors try to keep the manuscript as concise as possible.

Manuscript can be accepted in this present form.

Author Response

Thank you for your comments and your work which was very much appreciated!

Best Regards,

Birna

Reviewer 3 Report

The authors essentially did not provide an answer regarding the microbiological and chemical composition of colostrum. They simply state that the assays were not carried out. They should report it as methodological limitations of the article and of course discuss the planned future aspects. 

It should be mentioned under limitations

Author Response

Thank you for your valuable comments and your work which we appreciate highly!

We have added the following to the limitations of the study, in line 828: 

The main limitations of this study were the small sample size in each group "and limited analysis of the raw material such as basic chemical composition and no microbiological analysis performed."

In addition, we have added the following in the future perspectives section, in line 835: "Moreover, chemical and microbiological analysis are of importance as differences in the BC collection period may cause bioactive variability as well as any variation in processing, pasteurization, and storage conditions. Researching BC as a therapeutic agent for a medical condition, consistency of the product is vital."

Best Regards,

Birna